# What nature separated, and human joined together: About a spontaneous hybridization between two allopatric dogwood species (*Cornus controversa* and *C. alternifolia*)

**Barbara Gawrońska**[1]*, **Maria Morozowska**[2], **Katarzyna Nuc**[1], **Piotr Kosiński**[2,3], **Ryszard Słomski**[1]

**1** Department of Biochemistry and Biotechnology, Faculty of Agronomy and Bioengineering, Poznań University of Life Sciences, Dojazd, Poznań, Poland, **2** Department of Botany, Faculty of Horticulture and Landscape Architecture, Poznań University of Life Sciences, Wojska Polskiego, Poznań, Poland, **3** Institute of Dendrology, Polish Academy of Sciences, Parkowa, Kórnik, Poland

\* barb.gawronska@gmail.com

**Data Availability Statement:** Nuclear DNA sequences have been deposited and are available in the NCBI GenBank. accession no. MF770152 –

## Abstract

In this study, possible hybridization between two allopatric species, *Cornus controversa* and *Cornus alternifolia*, was explored using molecular and morphological approaches. Scanning electron microscope analyses of the adaxial and the abaxial leaf surfaces yielded a few new not yet described characters typical for the particular species and intermediate for hybrids. With the use of 14 Random Amplified Polymorphic DNA and 5 Amplified Fragment Length Polymorphism primer combinations, 44 fragments species specific to *C. controversa* and 51 species specific to *C. alternifolia* were obtained. Most of these bands were also found in putative hybrids. All clustering analyses based on binary data combined from both methods confirmed a separate and intermediate status of the hybrids. Hybrid index estimates for hybrids C1-C5 indicated that all were the first generation of offspring ($F_1$). Chloroplast intergenic spacers (*trn*F-*trn*L and *psb*C-*trn*S) were used to infer the hybridization direction. Based on the assumption of maternal inheritance of chloroplast DNA, *C. controversa* seems to be the maternal parent of the hybrid. Internal transcribed spacer sequences of the five hybrids analyzed here indicated higher similarity with the sequences of *C. controversa* (all shared the majority of its single nucleotide polymorphisms). Sequence analysis of *PI*-like genes fully confirmed the hybrid origin of C1-C5 hybrids. Our results also showed that two specimens in the *C. alternifolia* group, A1 and A3, are not free of introgression. They are probably repeated backcrosses toward *C. alternifolia*. Furthermore, molecular data seem to point not only to unidirectional introgression toward *C. controversa* (the presence of hybrids) but to bidirectional introgression as well, since the presence of markers specific for *C. controversa* in the profiles of *C. alternifolia* specimen A3 was observed.

MF770156; MF770162 – MF770164 and
MF783074 – MF783081.

**Funding:** This work was supported by the Ministry
of Research and High Education Research Capacity
Grants for Poznań University of Life Sciences, No.
506.181. 01.00 to BG, KN, RS and the Ministry of
Research and High Education Research Capacity
Grants for Poznań University of Life Sciences, No.
506.641.01.00 to MM, PK. The funders had no role
in study design, data collection and analysis,
decision to publish, or preparation of the
manuscript.

**Competing interests:** The authors have declared
that no competing interests exist.

## Introduction

Hybridization is an important factor in the evolution of plants, animals, and fungi [1–3]. Natural hybridization can be defined as the interbreeding of specimens from two distinct populations or groups of populations and is most easily recognized when formerly allopatric populations come into secondary contact. This sympatry often leads to the creation of a hybrid zone, with parental types, $F_1$ hybrids, and generation hybrids and backcrosses present in varying proportions [4]. In turn, introgression (or "introgressive hybridization") is described as the incorporation (usually via hybridization and backcrossing) of alleles from one species into the gene pool of a second one [5]. In many hybrid zones, differential introgression (alleles at some loci introgress more than others) was documented. Some of them tend to introgress easily, and this phenomenon has been described as "adaptive introgression" [6–8]. According to Harrison and Larson [4], boundaries between species are semi-permeable, which implies that differential introgression is the result of a selective process. Recent data indicate that permeability varies as a function of genome region. Thus, hybridizing taxa often remain distinct for only part of the genome. The proportion of the genome that is resistant to introgression varies among taxa and even across the same hybrid zone [9]. As mentioned above, introgression is strongly connected with hybridization and subsequent repeated backcrossing of hybrids with parental species. Such gene exchange can result in gene capture, a process in which a gene from a donor species is transferred irreversibly into a host species. Similarly, cytoplasmic gene flow between hybridizing taxa is referred to as cytoplasmic introgression. This process can eventually lead to cytoplasmic capture and the production of hybrids carrying the nuclear genome of a species and the cytoplasmic genome of another one [10,11].

Interspecific hybridization can also be an important step in plant speciation [12–16], leading to the origin of hybrid species in two different ways: via allopolyploidy [17,18] or homoploidy [13,16,19,20]. Homoploid hybrid speciation (speciation via hybridization without a change in chromosome number) is less common than polyploidy speciation. It is also a type of sympatric speciation, because the parental species must co-occur geographically to produce hybrids [19,21]. According to Feliner et al. [22], beyond this basic definition, there is controversy concerning the key aspects of the process, such as the relative proportions of each parental genome present in a hybrid species, the mechanisms leading to reproductive isolation, and its degree or the role played by hybridization in the process. The question is what evidence is required to demonstrate that hybrid speciation has occurred? In agreement with a number of previous [1,2,23] and recently published reviews [22,24], hybrid speciation is defined as a speciation event in which hybridization is crucial in the establishment of reproductive isolation. Schumer et al. [24] proposed three criteria that a putative hybrid should satisfy for confident consideration: (1) reproductive isolation of hybrid lineages from the parental species, (2) evidence of hybridization in the genome, and (3) evidence that this reproductive isolation is a consequence of hybridization.

The genus *Cornus* (Cornaceae) consists of about 60 species, which are mostly shrubs and small trees widely distributed in the temperate and subtropical regions of the Northern Hemisphere. The genus is known for its great morphological heterogeneity concerning mainly the structure of inflorescences and flowers as well as leaf morphology and leaf trichomes. Typical trichomes for *Cornus* species are unicellular, two-armed, with the arms oriented parallel to the epidermis (malpighian hair). Dogwoods express a wide variety of habits, from low herbaceous ground covers, such as *C. suecica* L., and multi-stemmed shrubs, such as *C. sericea* L., to small or large trees, such as *C. kousa* Buerger ex Miq. or *C. nuttallii* Audubon ex Torr. & A.Gray [25–29].

Phylogenetic relationships within the genus have been controversial for some time. Their complicated taxonomy resulted in many phylogenetic studies based on morphological and molecular data [25,29–35]. Additionally, the hybridization occurring between *Cornus* species frequently intensifies the confusion in the taxonomic treatment of the taxa involved. Hybridization may occur either between *Cornus* species with an overlapping distribution area growing in the wild or between species growing in close proximity in botanical collections. *Cornus ×arnoldiana* Rehder was the first recognized hybrid growing in cultivation in the Arnold Arboretum and was considered by Rehder [36] to be a hybrid of *C. amomum* subsp. *obliqua* (Raf.) J.S. Wilson and *C. racemosa* Lam., both species native to North America. This hybrid was later found in the wild from New England to Missouri, where its putative parents grow close to each other. Another example of hybridization between dogwood species was the case of *C. ×slavinii* Rehd., the hybrid of *C. rugosa* Lam. and *C. sericea* L. [= *C. alba* subsp. *stolonifera* (Michx.) Wangerin], grown in the Parks Department in Rochester, New York. This hybrid was also found growing wild from New York to Wisconsin within the range of its native parents [37]. The presence of some other *Cornus* hybrids growing in cultivation, such as *C. ×horseyi* Rehder (*C. amomum* Mill. × *C. macrophylla* Wall.) and *C. ×dunbarii* Rehder (*C. asperifolia* Michx. × *C. macrophylla*), or the natural hybrid between *C. racemosa* and *C. rugosa* from Michigan was also described [37,38]. These examples of hybridization concern several species representing the subgenus *Kraniopsis*, one of the two subgenera in the biggest group of blue-white fruited dogwoods (BW) [33].

To date, nothing is known about the hybridization between *C. alternifolia* L. f. and *C. controversa* Hemsl., two closely related and morphologically very similar dogwoods representing subgenus *Mesomora* from the BW group. Both are exceptional species in the genus; they have alternate leaves and a chromosome number of $n = 10$, and they are diploids [39]. Important differences between these two species include their habit, leaves, and fruit stone morphology. *C. alternifolia* is a shrub or small tree with branches that spread in irregular whorls, whereas *C. controversa* is a large tree up to 20 m tall with a distinct horizontal habit. The leaves of *C. alternifolia* differ from *C. controversa* in that there are fewer lateral leaf veins, and the leaf base, which is subrounded in the latter species, is cuneiform in shape. The fruit stones of both species are characterized by having a conspicuous cavity at the apex, which is larger at the *C. alternifolia* endocarps. Other differences concern the pubescence on the abaxial leaf surface and the shape and size of the inflorescence [25,27,28,40–42].

The species under study differ in their origin as *C. controversa* comes from East Asia, while *C. alternifolia* is from eastern North America (S1 Fig). However, according to Eyde [25], these two dogwoods originally had a wider distribution, as their endocarps were present in Tertiary fossil beds in Europe but are now extinct in these areas. According to Li [43] and Boufford and Spongberg [44], this resulted in the well-known disjunct distribution of these species between East Asia and North America. In turn, the little chloroplast DNA (cpDNA) variations between *C. alternifolia* and *C. controversa* found by Xiang et al. [45] suggest either that the divergence between the two species is relatively recent or that their cpDNAs have undergone a slower rate of evolution, or both. Such a close phylogenetic relationship leads to the question of whether hybridization is possible between these two species. The negative answer is easy when we eliminate natural hybridization, which is not possible since *C. alternifolia* and *C. controversa* are characterized by disjunct distribution, but hybridization might happen between specimens growing in close proximity in arboreta or other botanical collections. Such a case of spontaneous hybridization between two normally allopatrically distributed *Acer* species that were grown in a botanical garden was described by Liao et al. [46]. An example of hybridization between *Cornus mas* L. and *Cornus officinalis* Siebold & Zucc. from the Kórnik Arboretum was also described by Morozowska et al. [47].

Recently, several specimens of alternate-leaved *Cornus* species growing in two Polish botanical collections, in the Kórnik Arboretum and in the Adam Mickiewicz University Botanical Garden in Poznań, were observed according to their morphology. The preliminary morphological studies proved that at least two among all observed specimens growing in the Kórnik Arboretum, specified there as *C. controversa*, exhibit some divergences in their habit and leaf morphology with reference to their taxonomy [26,27,40]. Additionally, neither of these specimens flowered and fruited abundantly. Since the initial results of morphological observations indicated an intermediate character of the examined features typical either of *C. controversa* or *C. alternifolia*, we raised the hypothesis that these two specimens may be hybrids between two alternate dogwoods. With the aim to verify the taxonomic status of these two specimens, the preliminary macro- and micromorphological studies were performed with the help of the scanning electron microscope (SEM). To our surprise, the obtained results were not unambiguous—not only according to the two putative hybrids but also according to all other observed *C. controversa* specimens. That inclined us to verify, with the use of molecular methods, the hypothesis about the possible hybridization between two alternate leaved dogwoods growing in cultivation conditions. To get clear and reliable results concerning the origin of the examined specimens, we have included in our molecular and morphological studies several other specimens of *C. alternifolia* and *C. controversa* with documented natural origin as well as several herbarium specimens collected in natural sites. We also assumed that the detailed SEM micromorphological studies of leaf adaxial and abaxial surfaces performed on the material of natural origin will result in finding some new diagnostic features important in the taxonomy of both examined *Cornus* species.

Molecular data are widely considered to be better suited in hybridization research because they can more easily confirm hybridization than intermediate morphological trait values [16,23]. Three of the most commonly used methods are analysis of organelle genomes (cpDNA, mitochondrial DNA [mtDNA]), clustering-based methods that incorporate multilocus genotype data such as randomly amplified polymorphic DNA (RAPD) [48] or amplified fragment length polymorphism (AFLP) [49], and sequence comparisons of nuclear ribosomal DNA internal transcribed spacer (ITS) regions.

The use of the fingerprinting methods such as RAPD or AFLP allows one to assess the relationship and genetic distinction or lack of it between the putative hybrids and their progenitors. In turn, analyses of species-specific cpDNA and mtDNA enable the direction and intensity of hybridization to be defined. Moreover, the mechanism of uniparental inheritance of organelle genomes may be applied in studies on hybridization and introgression between closely related species. Potentially advantageous features of ITS sequences are their biparental inheritance compared to the uniparental inheritance of organellar DNAs and the assumed intragenomic uniformity due to the active homogenization of repeat copies, known as concerted evolution [50]. Increasingly, a fourth method using other low-copy genes is added to the methods presented above and often becomes crucial for the study. At any taxonomic level, if cpDNA and nuclear ribosomal DNA (nrDNA) phylogenies are poorly resolved, weakly supported, and/or incongruent with each other, the utility of other low-copy nuclear genes with rapidly evolving introns should be considered [51–53].

The MADS-box gene family encodes critical regulators determining floral development (formation of flowers, flowering time, and vegetative development in plants) [54]. Subsequent genetic analyses identified five different genes that provide floral homeotic functions (A, B, and C). All of these genes encode putative transcription factors [55]. *PISTILATA (PI)* is classified as a B-class gene of the MADS-box gene family and has already been investigated in the genus *Cornus* [56]. Moreover, studies conducted recently revealed that *PI* homologs exist as a single-copy gene in most of the species investigated, although in some species two or more copies were also found [57].

Two species, *C. alternifolia* and *C. controversa*, were thought to be closely related phylogenetically [33], but natural hybridization has never been observed in the field because of allopatry. However, theoretically, the two taxa growing together in any plant collection may spontaneously hybridize, and their progeny (hybrids) could reveal phenotypes that are in between the two parents. In such cases, the number of specimens of hybrid origin is usually small, and detection of hybrids is also more difficult. Moreover, we should also take into account the fact that we do not analyze cases occurring in the natural environment where there is the possibility of collecting material in a hybrid zone with parental type, $F_1$ hybrids, multiple generation hybrids, and backcrosses. Lack of access to a diverse pool of specimens of hybrid origin strongly limits the possibility of assessing the degree of introgression in both time and space. According to Hegarty and Hiscock [23], such studies will require the accumulation of evidence from multiple sources before definitive answers can be given.

In the present study, morphological characters and all most commonly used molecular methods mentioned above (fingerprinting, cpDNA restriction site variations, and sequence analysis of low-copy genes) were used to test the hypothesis that at least two specimens growing in the Kórnik Arboretum and characterized by intermediate morphology are hybrids of two allopatric *Cornus* species: *C. alternifolia* and *C. controversa*. Because introgression is a common consequence of hybridization, we also aimed to investigate if there is any evidence for introgression between putative parental species.

## Material and methods

### Plant material

Fresh and herbarium plant materials for morphological and molecular studies were collected from 9 specimens of C*ornus alternifolia*, 21 specimens of *C. controversa* (including putative hybrids between *C. controversa* × *C. alternifolia*), and additionally from 2 specimens of *C. macrophylla* (used as an out-group in some analyses). Cultivated specimens were growing in the Kórnik Arboretum, in the Adam Mickiewicz University Botanical Garden in Poznań, and in the Rogów Arboretum. Herbarium vouchers of the examined specimens were deposited in the Department of Botany, Poznań University of Life Sciences, Poland. Herbarium materials were obtained from the herbarium of the Institute of Dendrology in Kórnik (KOR; http://sweetgum.nybg.org/science/ih/). For details concerning examined specimens, see Table 1.

### Macro- and micromorphological studies

The preliminary macromorphological studies of leaves included the determination of an average number of lateral leaf veins and the shape of the lamina base. The observations were done on a sample of 10–30 leaves per individual, depending on the material availability.

Micromorphological observations were performed on the ultrastructure of the adaxial and abaxial leaf surface. Samples of fresh leaf fragments were air-dried under ambient conditions and coated with gold prior to observation. Micromorphological studies and photographic documentation were carried out with the use of a Zeiss EVO 40 scanning electron microscope. The observations were done on a sample of 5–10 leaves per individual, depending on the material availability. The terminology used to describe the leaves' ultrastructure follows Barthlott [58] and Barthlott et al. [59].

### Identification of hybrid origins based on molecular markers and sequence analysis

Because morphological data are likely to be unreliable in distinguishing $F_1$ hybrids from backcrosses and also may not indicate whether parents are free of introgression, a genetic analysis

**Table 1. List of *Cornus* individuals and herbarium specimens examined in morphological and molecular studies.**

| Sample code | Species | Herbarium or plant collection | Inventory code | Seed/seedling source |
|---|---|---|---|---|
| A1* | *C. alternifolia* | KA | 612 | Poland, AMU (from natural site, without precise location), planted in KA in 1929 |
| A2* | *C. alternifolia* | KA | 518/94 | Canada, Toronto (43˚ 04' N, 80˚ 10' W, 320 m), seeds from natural site, planted in KA in 1994 |
| A3* | *C. alternifolia* | AMU | P8XXX_3994 | Poland, AMU (from natural site, without precise location), planted in 1990 |
| A4* | *C. alternifolia* | RA | 8838a | Australia, Esperance Arboretum, 1974, collected in natural site |
| A5* | *C. alternifolia* | RA | 8838b | |
| A6* | *C. alternifolia* | RA | 8623 | Canada, Montreal Botanical Garden, Quebec, 1974, collected in natural site |
| A7* | *C. alternifolia* | RA | 8500 | Canada, Ontario, Wentworth Co., East Flamborough Township, Mountsberg, 1973, collected in natural site |
| A8* | *C. alternifolia* | RA | 8348 | Canada, Montreal Botanical Garden, Quebec, 1973, collected in natural site |
| A9 | *C. alternifolia* | KOR | 47458 | USA, North Dakota, Fargo, 1981, collected in natural site |
| C1* | *C. controversa* | KA | 611a | Poland, AMU (origin unknown), planted in KA in 1929 |
| C2* | *C. controversa* | KA | 611b | |
| C3* | *C. controversa* | KA | 611c | |
| C4* | *C. controversa* | KA | 611d | |
| C5* | *C. controversa* | KA | 8949_0826 | Poland, AMU (origin unknown), planted in KA in 1949 |
| C6* | *C. controversa* | RA | 15686/1 | Norway, Arboretum and Botanical Garden Hjellestad, University of Bergen, 2002; collected in natural site in Honshu, Daisen, Japan |
| C7 | *C. controversa* | RA | 15686/2 | |
| C8* | *C. controversa* | RA | 15686/3 | |
| C9* | *C. controversa* | RA | 15686/4 | |
| C10 | *C. controversa* | RA | 15686/5 | |
| C11* | *C. controversa* | RA | 15686/6 | |
| C12 | *C. controversa* | RA | 15686/7 | |
| C13* | *C. controversa* | RA | 15686/8 | |
| C14* | *C. controversa* | RA | 15686/9 | |
| C15* | *C. controversa* | RA | 15686/10 | |
| C16* | *C. controversa* | RA | 15223/1 | South Korea, Seul Arboretum, Gangwon Province, 2000, collected in natural site |
| C17* | *C. controversa* | RA | 15223/2 | |
| C18 | *C. controversa* | KOR | 12943 | North Korea, Diamond Mountains, Kumgang-san, W of Kosong, 1978, collected in natural site |
| C19 | *C. controversa* | KOR | 12947 | North Korea, Myohyang-san, E from Hangsan, 1980, collected in natural site |
| C20 | *C. controversa* | KOR | 44134 | |
| C21 | *C. controversa* | KOR | 44133 | |

*(Continued)*

**Table 1.** (Continued)

| Sample code | Species | Herbarium or plant collection | Inventory code | Seed/seedling source |
|---|---|---|---|---|
| **C22**[*] | *C. macrophylla* | RA | 14226a | Japan, Kyoto Botanical Garden, Kyoto Forest University, Kitayama, 1995, collected in natural site |
| **C23**[*] | *C. macrophylla* | RA | 14226b | |

AMU: Botanical Garden of Adam Mickiewicz University in Poznań, Poland (52˚ 25' N, 16˚ 53' E); KA: Kórnik Arboretum, Institute of Dendrology, Polish Academy of Sciences, Poland (52˚ 14' N, 17˚ 05' E); KOR: Herbarium of Institute of Dendrology in Kórnik, Polish Academy of Sciences, Poland; RA: Rogów Arboretum, Warsaw University of Life Sciences–SGGW, Poland (51˚ 49' N, 19˚ 53' E).

[*] specimens examined with molecular markers; putative hybrid specimens are marked in bold

of 24 plants was performed using two methods: RAPD [60] and AFLP [61]. Total genomic DNA was extracted from fresh young leaves of all specimens indicated in Table 1 (including *C. macrophylla* as an out-group) using a modified CTAB (cetyltrimethyl ammonium bromide) method [62] previously described [47]. The contaminating RNA was removed by digestion with RNase A. DNA quality and concentration were estimated by electrophoresis and spectrophotometry, adjusted to 20 ng/μl, and used as a template in polymerase chain reactions (PCR).

## RAPD analysis

PCR RAPD amplification was performed in volumes of 25 μl containing a double concentrated, ready-to-use PCR master mix (Thermo Fisher Scientific), a 20-pM decamer primer, and 50 ng of template DNA. Detailed methods for the preparation and program of PCR reactions are provided in Morozowska et al. [47]. In order to check the reproducibility of the RAPD markers, some of the RAPD primers were tested two to three times. Most of them were also checked using DNA from different isolations as a template. In all cases, the primers showed reproducible results. The amplified products were separated in 8% polyacrylamide gels in 0.5 x TBE buffer in the presence of size markers, silver-stained, and photographed.

For preliminary selection, 52 decanucleotide primers (kits A, B, and H; Operon Technologies Inc., USA) were probed using a subset of the parental DNA samples. The 14 primers that resulted in polymorphic bands unique to each parental species were used in further reactions with all examined species and listed in Table A in S1 Table.

## AFLP analysis

DNA templates for AFLP reaction were prepared by digesting 200 ng of genomic DNA with *Eco*RI and *Mse*I restriction enzymes (37˚C, 3 h), and the fragments were then ligated to *Eco*RI/*Mse*I adapters. The ligation mixture was diluted 5-fold with sterile distilled water and used as a template for the preamplification using 26 PCR cycles of 94˚C, 56˚C, and 72˚C each for 60 s and primers *Eco*RI/*Mse*I (+1/+1) in a PTC-100 thermocycler (MJ Research). Sequences of AFLP adapters and primers were based on Vos et al. [61]. Preselective amplifications were performed in a 25 μl volume of 7.5 pmol of each primer, 2.5 μl of 10x PCR buffer, 1.2 μl of 25 mM $MgCl_2$, 1.0 μl of 5 mM dNTPs, 1 U *Taq* DNA polymerase (Sigma), and 5 μl of template. Part of the preamplification mixtures were confirmed in agarose gels, and the rest of the reactions were diluted 1:10 in $dH_2O$ and used as a template for selective amplification. Selective amplification was done in a total volume of 25 μl consisting of the same ingredients as above (apart the 5 pmol of FAM-labeled *Eco*RI primer and 25 pmol of non-labeled *Mse*I primer) and 5 μl of diluted template. Selective PCR was done using the program that consisted of two steps: 13

cycles at 94◦C for 30 s, 65◦C for 30 s (touch down of 0.7◦C per cycle), and 72◦C for 120 s followed by 23 cycles at 94◦C for 30 s, 56◦C for 30 s, 72◦C for 120 s, and a final extension of 72◦C for 10 min. Each reaction was repeated at least two times. Amplification products were separated and detected on an ABI Prism 3500 capillary sequencer. GeneScan 600 LIZ-labeled size standard (Applied Biosystems) was used for fragment sizing. The fluorescent AFLP patterns were scored using GeneMapper version 3.7 (Applied Biosystems). Five of the selective *Eco*RI/*Mse*I (+3/+3) primer pairs (E-AAG/M-CAC, E-AAG/M-CTC, E-AAG/M-CTG, E-AAG/M-CAG, and E-ACA/M-CTG) were tested to find specific diagnostic markers for each parental species.

## RAPD and AFLP data analysis

The total number of bands and the distribution of bands across taxa as well as the number of polymorphic bands, average number of bands per primer, and bands shared among species were examined. To estimate the degree of gene flow, some categories for marker bands were used in these calculations. According to Delaporte et al. [63] monomorphic bands were those present in all specimens under the study, polymorphic were those bands present in at least one but not all specimens, and unique bands were those present in at least one individual in a taxon and not present in any other. For qualitative identification of hybrid specimens, diagnostic markers (present in one species and not present in the other species) were also defined [64]. Those diagnostic markers (a subset of all unique markers, always present in a species) are known as species-specific.

Bands (peaks) were scored as dominant markers. Bands that were present were given a value of 1, and absent bands were given a value of 0. The binary matrices of AFLP and RAPD phenotypes were the basis for calculation of Dice similarity [65]. Genetic similarity was transformed into genetic distance using the formula $Dij = 1–Sij$. The Dice matrices were used subsequently for the construction of the unrooted neighbor-joining (NJ) dendrogram and phylogenetic network based on the Neighbor-Net algorithm using SplitsTree4 v.4.13.1 software [66]. Neighbor-Net [67] is better suited to represent multiple phylogenetic processes, including hybridization, and to visualize reticulate relationships among individuals. The above-mentioned analyses were supplemented by the Unweighted Pair Group Method with Arithmetic Mean (UPGMA) dendrogram and a principal coordinate analysis (PCoA), both performed in PAST 3.15 [68]. Bootstrapping was used to calculate a support value for each node on the dendrograms (1000 replicates).

To identify genetic groups without a priori knowledge of sample origin, we used a model-based Bayesian approach implemented in the program STRUCTURE, version 2.3.4 [69]. A STRUCTURE analysis was performed based on an admixture model, with the recessive alleles option set to 1, as AFLPs and RAPDs are dominant markers [70]. Ten independent repetitions for each number of groups (K) ranging from one to five were performed with a burn-in of $10^5$ steps, followed by $2 \times 10^5$ MCMC iterations. The alignment of the results across 10 replicates of analyses was assessed using CLUMPAK [71], and the best number of clusters was determined according to Evanno's ΔK method [72] as implemented in the online program STRUCTURE HARVESTER [73].

Finally, for putative hybrids, hybrid indices were calculated using diagnostic markers (from both analyses). Each plant was scored for presence or absence of each parental marker. This index is useful to identify intermediate individuals and show backcrosses. According Fritz et al. [74], hybrid index values (distances from *C. alternifolia*) were calculated as follows: "pure" *C. alternifolia* were given a score of 0; the presence of any *C. controversa* marker or the lack of any *C. alternifolia* marker increased the index value up to a maximum of 1 for "pure" *C.*

*controversa*. Theoretically F$_1$ plants would have an index of 0.5 and possess all markers from both parents. Backcrosses are expected to lack a portion of the markers from one species. Lastly, 22 specimens were tested using 14 RAPD primers (21 species-specific markers for *C. alternifolia* and 10 species-specific for *C. controversa*) and 5 selective *Eco*RI/*Mse*I (+3/+3) primer pairs (30 species-specific markers for *C. alternifolia* and 34 species-specific markers for *C. controversa*). Unique markers that were polymorphic within species were not used in this analysis.

The reliability of manual calculations has been confirmed using computer programs. The program INTROGRESS [75], as implemented in the R programming environment [76], was used to calculate a hybrid index for each individual based on the combined RAPD and AFLP binary matrix. Specimens of *C. alternifolia* and *C. controversa* with documented native origin were chosen as the parental groups.

## Chloroplast DNA analysis

PCR-RFLP combines both the PCR and RFLP techniques, which has the ability to discriminate between genotypes based upon the presence or absence of restriction sites within the amplified DNA. Two fragments of plastid DNA were amplified, including the non-coding region between *trn*F and *trn*L genes of cpDNA and CS region *psb*C and *trn*S. The first fragment was amplified using universal primers described in Taberlet et al. [77], while the CS region was amplified by PCR with pairs of universal primers described in Demesure et al. [78]. PCR reactions were carried out in a total volume of 25 μl containing a ready-to-use PCR master mix (Thermo Fisher Scientific), 20 pmol each of primer, and about 40 ng of template DNA using a PTC-100 thermal cycler (MJ Research), and the program consisted of one cycle of preliminary denaturation of DNA for 5 min at 95˚C, 30 cycles of denaturation for 45 s at 92˚C, primer annealing for 45 s at 53˚C for *trn*F-*trn*L or 58˚C for *psb*C- *trn*S and 2–4 min at 72˚C (depending on size of the fragment), followed by one cycle of 10 min at 72˚C. Efficiency of amplification was verified by the electrophoretic separation of 5 μl of PCR products in 1.5% agarose gel.

PCR-RFLP analyses of the amplified cpDNA regions were performed with the use of restriction enzymes. The enzymes applied for digestion of each region were derived in a random selection. 5 μl of PCR product underwent digestion (2–4 h with 1–2 units of restriction enzyme, reaction volume of 10 μl), following the reaction conditions recommended by the enzyme provider (Thermo Fisher Scientific). Both regions were subjected to restriction analysis using *Alu*I, *Dra*I, *Hind*III, *Hinf*I, *Mbo*I, *Mse*I, *Msp*I, *Pst*I, *Taq*I, and *Xba*I enzymes. Restriction fragments were separated on 8% polyacrylamide gels. After electrophoresis, silver staining was carried out using the standard procedure.

## ITS and Cor PI cloning and sequencing

Amplifications of the nrDNA ITS region and *Cor PI* loci were performed on a PTC-100 thermal cycler (MJ Research) programmed with an initial denaturing at 95˚C for 2–4 minutes, followed by 25 (30) cycles of 95˚C for 30 s, 55(50)˚C for 30 s, and 72˚C for 1 min, with a final extension at 72˚C for 10 min, for both regions respectively. The total reaction volume was 25 μl, containing 2 × PCR Master Mix (Thermo Fisher Scientific), 10 pmol of each primer, and 50 ng of template DNA. The use of universal ITS4 and ITS5 primers developed by White et al. [79] resulted in inefficient amplification, and new primer sets flanking this region had to be constructed. These primers (forward primer: ITSFBG 5'–GCG GAA GGA TCA TTG TCG AAA CCT GC–3' and reverse primer: ITSRBG 5'–GTC GCG GTC GAT GCG CCG AG–3') were designed based on the sequence of *C. controversa* deposited in GenBank (ID: JF980315). Similarly, on the basis of sequences available in GenBank (ID EU447709), a pair of

species-specific primers (forward primer: CP1F1 5'-GCA TGA GTA CTG CAG CCC TG-3' and reverse primer: CP1R1 5'-GAT CTG TTC AAA CAA TTC ATG TTG-3') was designed and used for amplification of an approximately 500-bp fragment of the *Cornus*-specific *PI*-like gene.

Because direct sequencing of PCR fragments produced the superimposed chromatograms on multiple sites and unreadable peaks were observed, cloning was performed to isolate the different types of sequences for each individual. Ligation reactions were conducted following the manufacturer's protocol with a pGEM-T Easy Vector System from Promega. Positive transformants were detected by the insert size through PCR screening using M13 5'-GTA AAA CGA CGG CCA G-3' forward and M13 5'-CAG GAA ACA GCT ATG AC-3' reverse primers. Two to three positive clones representing each banding pattern (PCR products after gel electrophoresis) were directly sequenced in both directions. Forward and reverse sequencing was performed using the same forward and reverse primers. The nucleotide sequence data obtained were then compared with data available from GenBank using the FASTA program [80]. Multiple sequence alignments were created by the Vector NTI program (version 5.0) using the Clustal W algorithm [81].

## Results

### Morphological analysis

According to the obtained results of the preliminary morphological analysis, the average number of the leaf lateral veins of all the *C. alternifolia* examined specimens was between 5.0 and 6.1. The average number of leaf lateral veins of all the *C. controversa* specimens examined, including the two putative hybrid specimens C3 and C4, was between 7.0 and 7.6.

The shape of the lamina base was cuneiform in 75.0%-82.0% of the leaves of all the examined *C. alternifolia* specimens. With the exception of the two putative hybrid specimens, 40.0%-90.0% of the leaves of the examined *C. controversa* specimens were characterized by the subrounded shape of the lamina base. According to putative hybrids C3 and C4, the subrounded lamina base was observed in approximately 24.0% of the examined leaves.

According to the results of SEM studies, the microornamentation pattern of the adaxial lamina surface in leaves of all *C. alternifolia* and *C. controversa* specimens examined, including the two putative hybrid specimens, was reticulate with a striated cuticle. Cuticle striations were either parallel with each other, or they were more or less wavy. Anticlinal cell walls of epidermal cells on *C. alternifolia* leaves were often raised and thus distinctly visible; however, sometimes they were flat. Anticlinal cell walls of epidermal cells on *C. controversa* leaves were mostly flat and not well visible underneath the cuticle striations, but sometimes they were raised. On the adaxial leaf surfaces of all *C. alternifolia* and *C. controversa* specimens examined, the thin and smooth layer of epicuticular waxes and delicate wax plates were present (S2 Fig). On leaves of both examined *Cornus* species, straight pseudo-filiform trichomes were present along the major and secondary veins, especially close to the lamina base. A single filiform-like arm of such a trichome was long and flexible, sometimes twisted, with micro-striae. The same kind of trichomes were seldom observed in the islets between the veins close to the apex and on the margins of the lamina.

The microornamentation pattern of the abaxial lamina surface of both species under study was reticulate, and it was covered by the strongly and profusely rippled cuticle. However, the type of the cuticular pattern was distinctly different for both studied *Cornus* species. The pattern on the *C. alternifolia* leaves was papillose-striated, while the pattern on the *C. controversa* leaves was papillose-coronulate. Between the cuticular papillae projections and cuticle striations, the numerous anomocytic stomata were scattered on leaves of both examined species

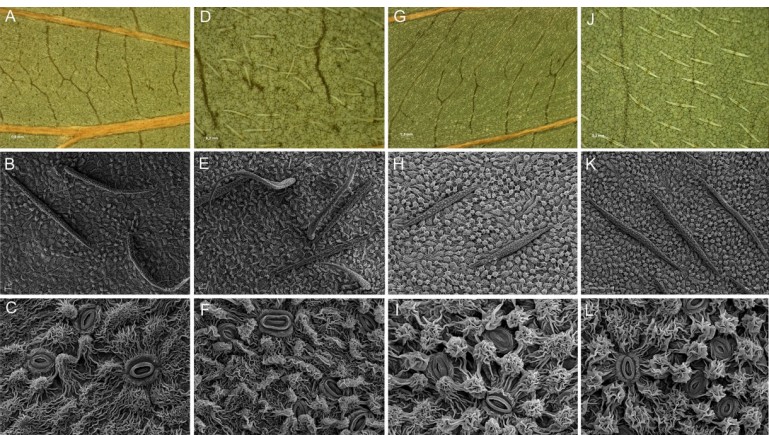

**Fig 1. LM figures and SEM micrographs of the abaxial leaf surface of *Cornus alternifolia* and *C. controversa*.** *C. alternifolia*: specimens A9 (A-C) and A2 (D-F); *C. controversa*: specimens C18 (G-I) and C9 (J-L) showing the reticulate microornamentation pattern with papillose-striated (*C. alternifolia*) (B, C, E, F) and papillose-coronulate (*C. controversa*) (H, I, K, L) cuticle pattern with pseudo-filiform one-armed or flat-symmetrical two-armed trichomes, respectively. SEM magnification 550, 700 and 3000; specimen symbols as in Table 1.

(Fig 1). The papillose-striated cuticle pattern was characterized by the presence of rounded or elongated dome-shaped papillae that did not protrude very much. Rounded papillae were rather rare; usually they did not occur in each epidermal cell, and their frequency was quite variable. These papillae were covered by secondary ornamentation of irregular or radiating striae. The elongated papillae usually occupied the surface of two to three epidermal cells, and they were covered with profusely rippled cuticle. The rest of the periclinal surface of the epidermal cells, between the papillae, was thickly covered by the wavy cuticle striations. The stomata were surrounded either by rounded or elongated papillae and by radiating cuticle striations, or by both of these cuticle formations (Fig 1 B, 1C, 1E and 1F). The papillose-coronulate cuticle pattern was characterized by the presence of rounded, dome-shaped, strongly protruding striate papillae, occurring one per cell. Cuticle striations present on such papillae often formed buttresses around each papilla. These papillae were interconnected by the cuticle folds that joined them and take the form of narrow cuticular ridges or ropes. On the periclinal surface of the epidermal cells around the papillae, the delicate, wavy, and low-protruding cuticle striations were present. The stomata were always surrounded by few rounded striate papillae (Fig 1H, 1I, 1K and 1L).

Both species under study also differed according to the type of the trichomes present on the abaxial leaf surface. On the *C. alternifolia* leaves, most of the trichomes were pseudo-filiform, with a single long flexible and sometimes wavy arm, and of irregular orientation (Fig 1A, 1B, 1D and 1E). On the *C. controversa* leaves, the numerous flat-symmetrical trichomes had two stiff, equal-length arms attached to the epidermis. The observed trichomes were parallel to the secondary veins and parallel to each other (Fig 1G, 1H, 1J and 1K). The surface of the trichomes described above had micro-papillae or sometimes micro-striae.

The papillose-striated cuticular pattern described above was typical for all of the *C. alternifolia* specimens examined. With reference to *C. controversa*, the described papillose-coronulate cuticular pattern was typical for all of the specimens with the documented natural origin. However, with reference to both putative hybrid specimens C3 and C4 as well as to the three other *C. controversa* specimens C1, C2, and C5 of unknown origin, the cuticular microornamentation pattern of the abaxial leaf surface was characterized by several differences in comparison with the papillose-coronulate cuticular pattern observed on the leaves of the *C.*

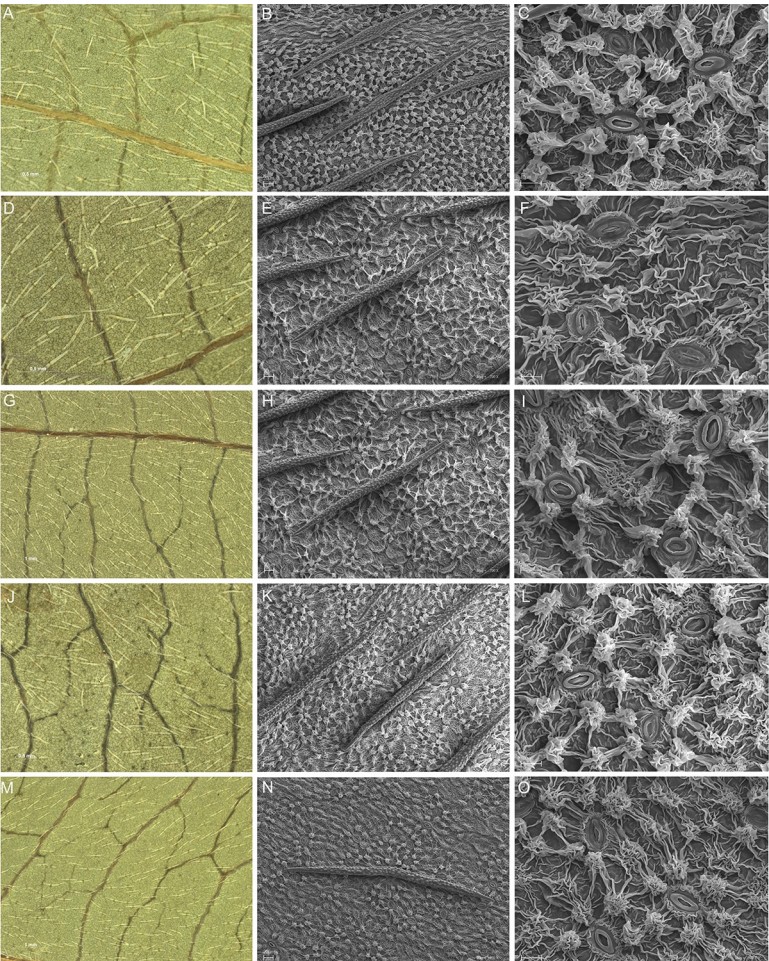

**Fig 2. LM figures and SEM micrographs of the abaxial leaf surface of putative hybrid specimens of *Cornus alternifolia* and *C. controversa*.** Specimens C1 (A-C), C2 (D-F), C3 (G-I), C4 (J-L) and C5 (M-O) showing: flat, sometimes elongated papillae not always present in each cuticle cell (E, F, H, I, N, O); the periclinal surface between the papillae thickly covered by the wavy cuticle striations (F, I, L, O); the presence of both two- and one-armed trichomes (A, D, J). SEM magnification 700 and 3000; specimen symbols as in Table 1.

*controversa* specimens with documented natural origin. On the abaxial leaf surface of the leaves of the *C. controversa* C2, C3 and C5 specimens, the papillae were rather flat and not always present in each cuticle cell. (Fig 2E, 2F, 2H, 2I, 2N and 2O). For both the C3 and C4 putative hybrid specimens, as well as for C2 and C5 specimens, in the places where there were no papillae, the periclinal walls of the cuticle cells were thickly covered by the wavy cuticle striations (Fig 2F, 2I, 2L and 2O). On the abaxial leaf surface of *C. controversa* C1, C2 and C4 specimens, both the flat-symmetrical two-armed trichomes and the pseudo-filiform one-armed trichomes were present, and the trichomes were not always parallel to each other (Fig 2A, 2D and 2J).

## RAPD and AFLP analyses

Fourteen of the fifty-two initial primers produced clear and reproducible polymorphic bands among the 24 genotypes. Those random primers generated a total of 352 RAPD bands, of which 43 were typical of *C. macrophylla*. Among the remaining 309 bands (without *C.*

*macrophylla* individuals), 14 fragments (4.5%) were monomorphic and 295 (95.5%) were polymorphic. The size of amplified fragments ranged between 160 and 1300 bp for all primers. A total of 352 loci (bands), or a mean of about 25.1 bands (24.1 polymorphic bands) per primer, were generated. RAPD analysis with 14 PCR primers produced banding patterns that were conserved within each species with a number of bands unique to *C. controversa* (53 bands) and *C. alternifolia* (68 bands). The results of RAPD analysis are summarized in Table A in S1 Table.

Five AFLP primer combinations yielded a total of 1000 amplification products including those characteristic only for out-group (115 bands). In the main studied group, 832 out of 885 (94%) were polymorphic across all 22 accessions (without *C. macrophylla* specimens). The number of polymorphic bands per primer combination ranged from 158 to 212, with an average of 189 bands. The percentage of polymorphic bands was variable and dependent on the primer combination used. The highest percentage of AFLP polymorphisms was obtained for *Eco*AAG/*Mse*CAC and *Eco*ACA/*Mse*CTG combinations, where 199 out of 205 and 158 out of 163 (97.0%), respectively, bands were polymorphic. Moreover, 267 (32.09%) of the total amplified polymorphic DNA bands distinguished the two parents. The two taxa, *C. controversa* and *C. alternifolia*, had 116 and 151 unique bands, respectively, and both sets of these specific bands were observed in hybrids. For details see Table B in S1 Table.

Cluster analysis of the dogwood genotypes was performed based on data from polymorphic RAPD and AFLP bands (including specimens from the out-group), and the estimated similarity coefficient was based on a combination of 1352 markers. Pairwise values of Dice's coefficient similarity between all possible pairs of genotypes ranged from 0.292 for C13 and M1 to 0.868 between two *C. macrophylla*. Within the main group, the highest genetic similarity (0.859) was observed between *C. alternifolia* A4 and A5, and the lowest values of genetic similarity (0.434–0.437) were between *C. alternifolia* (A1) and *C. controversa* (C6, C8) accessions.

The UPGMA dendrogram constructed on the Dice coefficient similarity matrix showed that all genotypes could be resolved into four main groups: three of them corresponded to the analyzed species: *C. macrophylla* (out-group), *C. alternifolia*, and *C. controversa*. Nevertheless, the specimens C1-C5 were distinguished as a separated subgroup in the clade of *C. controversa* (S3 Fig). The Neighbor-Net network (Fig 3) and the neighbor-joining unrooted tree (S4 Fig) also confirmed, to an even greater extent, the separateness of specimens C1-C5, which formed their own clade lying between clades of *C. alternifolia* and *C. controversa* and consisted of specimens with documented native origin. Similar patterns emerged from PCoA, which divided all compared specimens into four groups in the space between the two first coordinates; the first two components of the principal coordinates accounted for 58% of the total variation. Three distinctly isolated groups encompassed individuals of the analyzed dogwood species (including *C. macrophylla* as a referential group). And again, the fourth separate group consisted of accessions C1-C5, which were located between the group of *C. alternifolia* and the group encompassing the remaining samples of *C. controversa* (Fig 4). The distances among clusters were conspicuous as compared to the distances among specimens within the groups.

The Bayesian clustering analysis revealed that the best supported number of clusters was K = 2. The observed pattern of differentiation corresponded to the division of the samples into three groups. Two of them were homogenous and consisted of specimens assigned entirely to a single cluster (samples of *C. controversa*, C6, C8, C9, C11, and C13-C17, to cluster 1; samples of *C. alternifolia*, A1-A8, to cluster 2). The third group (samples C1-C5) showed a distinct overlap of the gene pools of the two previous groups (Fig 5A). Meirmans [82] suggested that all biologically relevant K-numbers, not only the best one, should be interpreted, as alternative numbers can reveal some interesting details. The STRUCTURE analysis that assumed the higher number of clusters K = 3 gave a similar picture, but samples C1-C5 were almost entirely

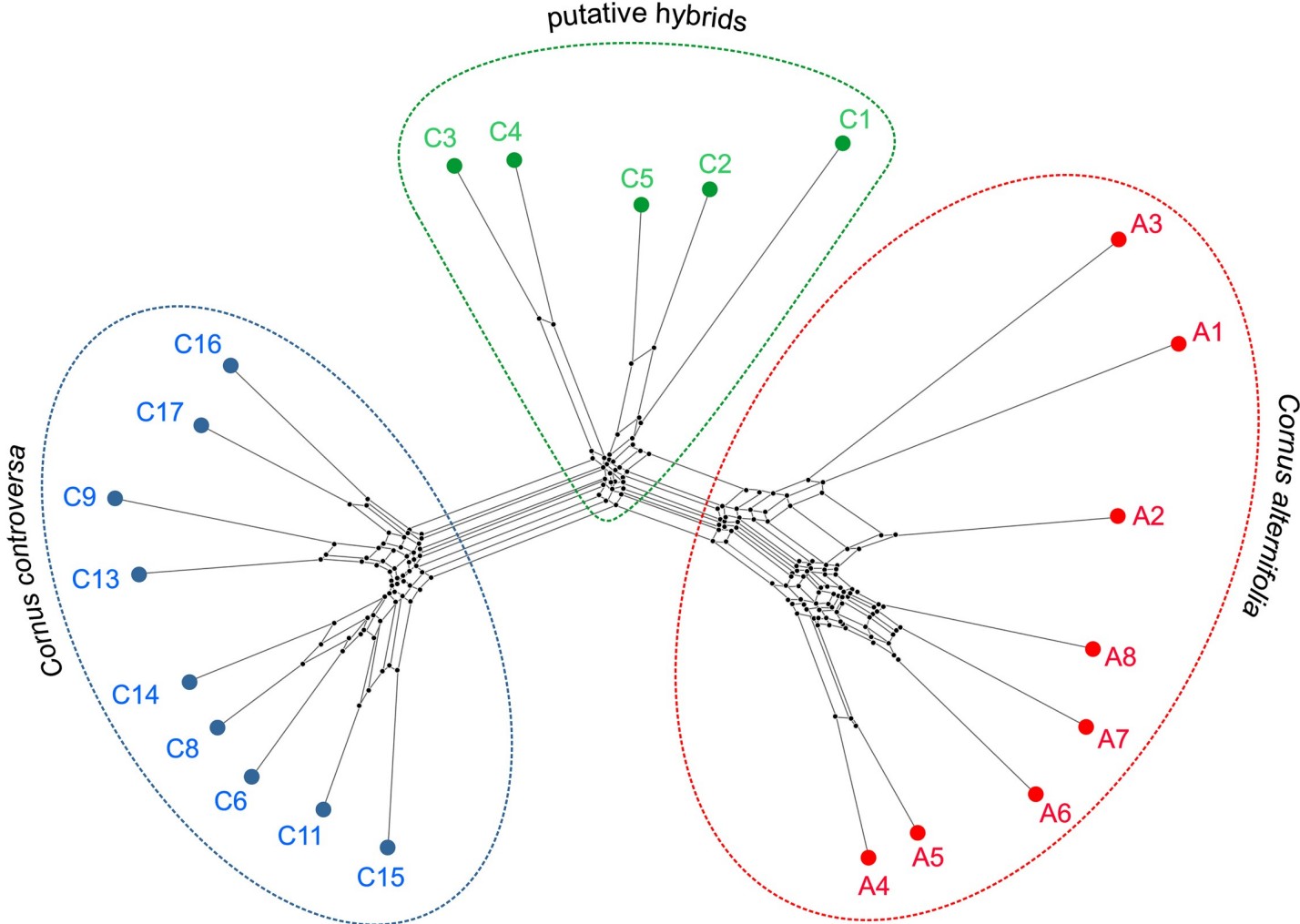

**Fig 3. Neighbor-Net derived from combined RAPD and AFLP binary matrices of 22 individuals representing *Cornus alternifolia*, *C. controversa* and their putative hybrids.** Accessions codes as in Table 1.

assigned to a separate "hybrid" cluster (Fig 5B). It is worth noting that this third "hybrid" cluster was also present in some specimens of *C. alternifolia* (samples A1 and A3, with a proportion of the memberships ranging from 19% to 27%, respectively).

Analysis of RAPD and AFLP results confirmed preliminary assumptions related to the presence of two potential hybrids (C3 and C4) in the study group. In addition, it showed that for the next three individuals described as *C. controversa* (C1, C2, and C5), there is also a high probability of hybrid origin. Almost all of the DNA bands amplified from the five putative hybrids co-occurred in either *C. controversa* or *C. alternifolia* banding patterns. Most of the 388 DNA bands (both analyses) that differentiated *C. controversa* and *C. alternifolia* were also present in the putative hybrids. Apart from these, some bands unique to hybrids were also present in their profiles. Such bands (71) have occurred both in individual cases as well as in several or even all five hybrids (one fragment in OPH-3 and in AAG/CAC, AAG/CAG combinations and four such bands in AAG/CTG combination). Moreover, all these values generated for the five putative hybrids seem to indicate that they are approximately equally related to each parent.

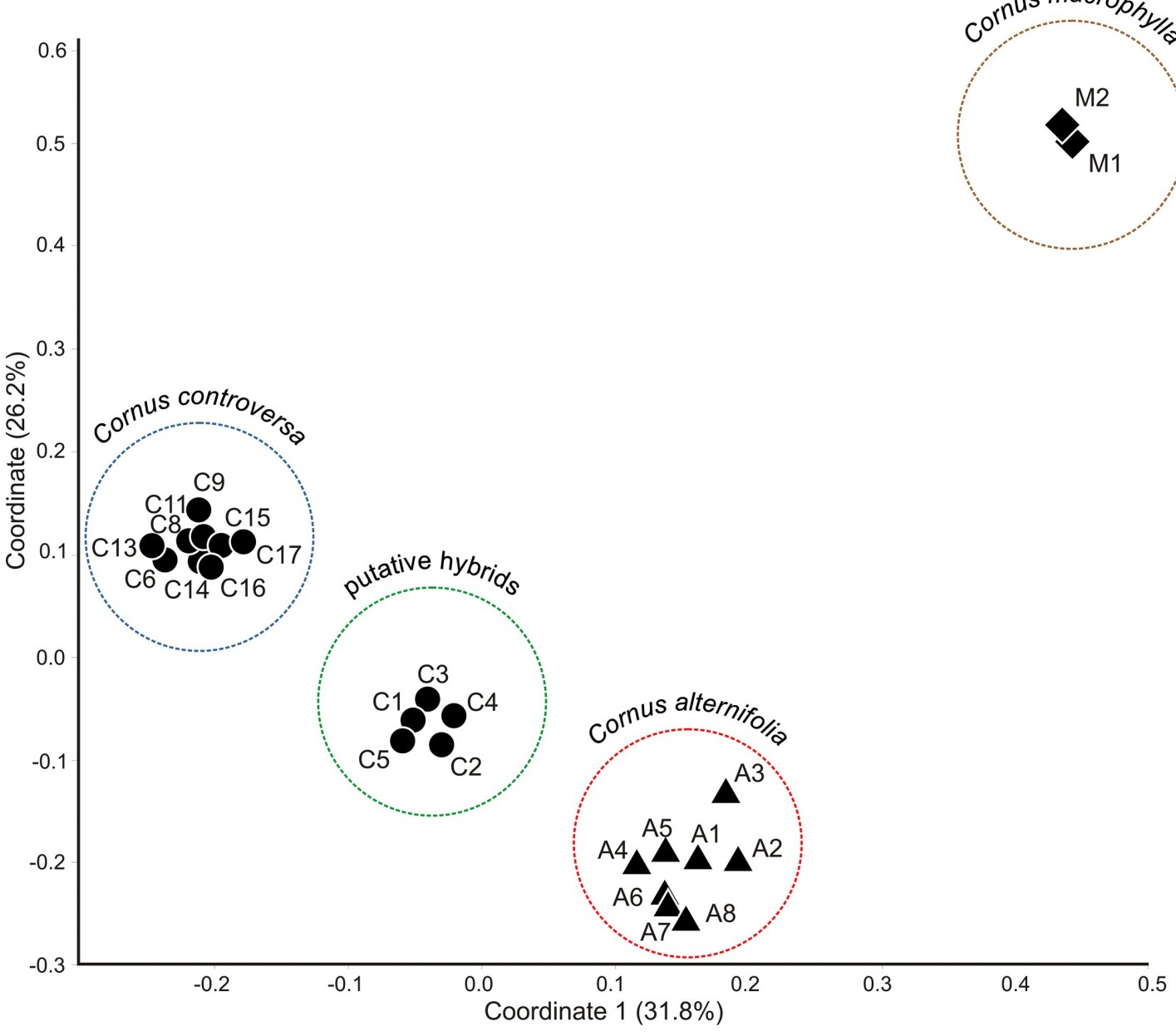

**Fig 4. Scatter plot of PCoA of studied *Cornus alternifolia*, *C. controversa*, and *C. macrophylla* (outgroup) accessions, based on Dice distances calculated from RAPD and AFLP combined binary matrices.** Accessions codes as in Table 1; two first coordinates explain 58% of the total observed variation.

To estimate the genetic status of *C. controversa* and *C. alternifolia* plants morphologically identified as "pure" and putative hybrids, markers unique to each parent were identified. As mentioned above, markers that were polymorphic within species were not used in this analysis. According to Fritz et al. [74], such markers may limit their usefulness in quantifying the genetic composition of putative hybrids. For further analysis, 95 unambiguous marker loci, 44 present in "pure" *C. controversa* and absent in "pure" *C. alternifolia* and 51 present in "pure" *C. alternifolia* and absent in "pure" *C. controversa*, were chosen. Based on these markers, for eight plants from the Kórnik Arboretum and Botanical Garden in Poznań (including five

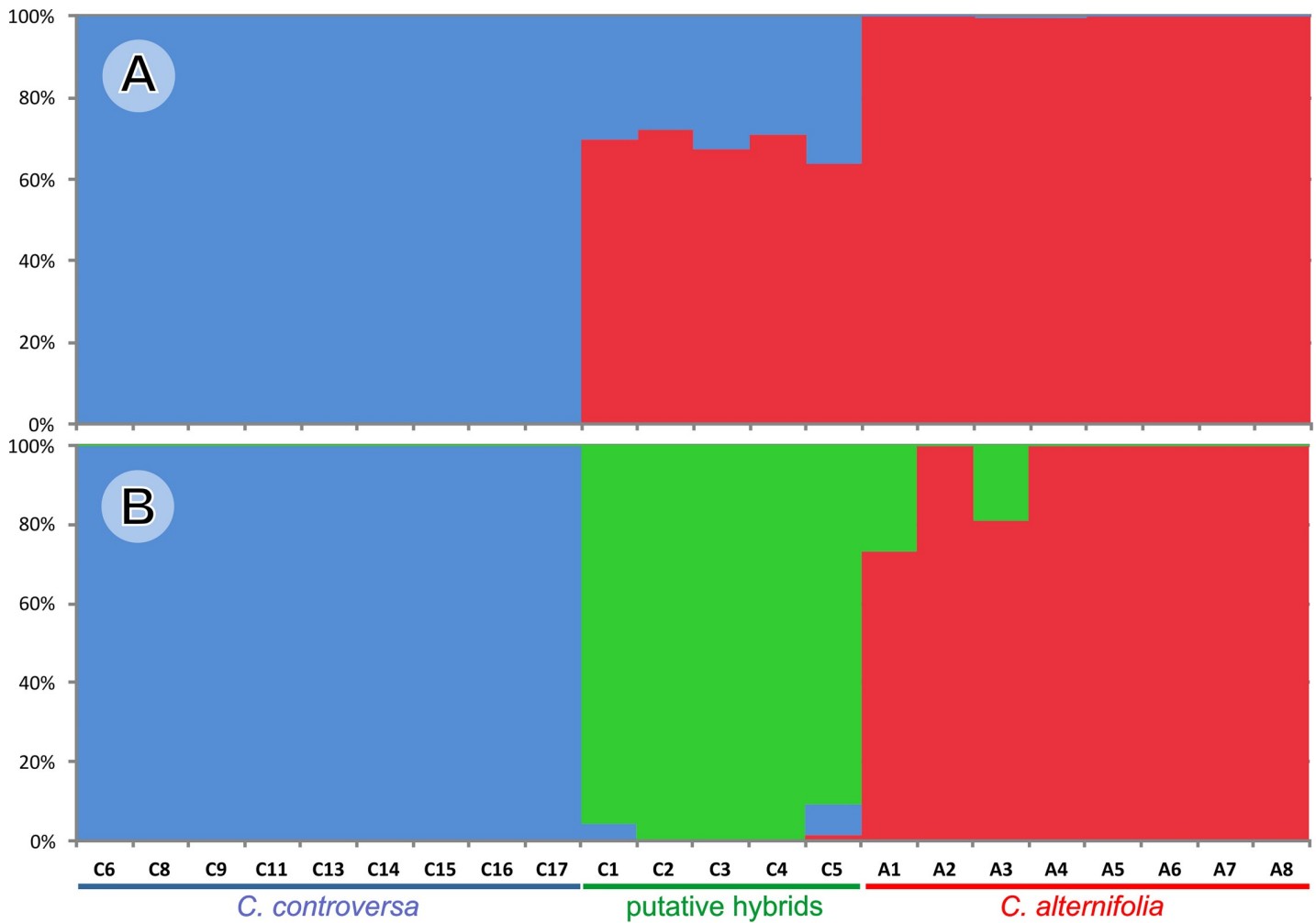

**Fig 5. Results of Bayesian analysis (STRUCTURE) of 22 samples of *Cornus alternifolia*, *C. controversa* and their putative hybrids, for K = 2 (A) and K = 3 (B).** Each column represents a different individual and the colors represent the probability membership coefficients (accessions codes as in Table 1).

putative hybrids and three *C. alternifolia* specimens A1-A3), a hybrid index was calculated. Each plant was scored for the presence or absence of each parental marker. The results are summarized in Fig 6.

Based on the hybrid index, both analyses corroborated the field identification of three plants as "pure" or almost "pure" *C. alternifolia* (A1, A2, and A3). Two of them differ only in lacking 15 (A1) and 9 (A2) of the 51 *C. alternifolia* markers, and neither had *C. controversa* markers (a distance from *C. alternifolia* of 0.16 and 0.09, respectively). The A3 individual is noteworthy because although it differs only in lacking 10 of the 51 *C. alternifolia* markers, it also has four *C. controversa* markers. This plant has a distance of 0.15 from *C. alternifolia* and is here interpreted as *C. alternifolia* (at least on the basis RAPD and AFLP). Three other plants (C1, C2, and C5) were identified as *C. controversa* in the field; these plants have morphological characters similar to *C. controversa* but appear to be F$_1$ based on RAPD/AFLP analyses. For further analyses, these three plants together with the putative hybrids C3 and C4 are treated as hybrids.

Among the five putative hybrids, there were two plants (C2, C5) that exhibited perfect marker additivity expected in an F$_1$ (index = 0.5) and one plant (C3) that deviated by only one

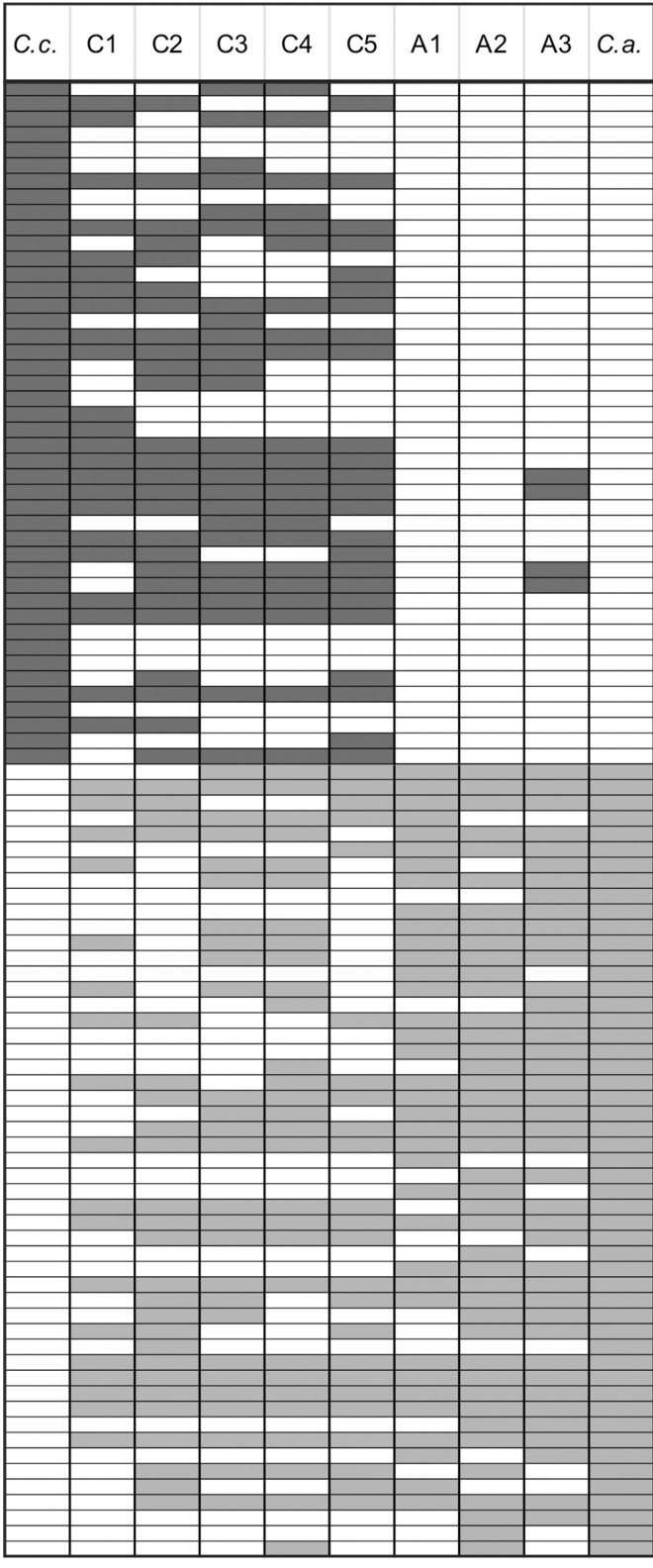

■ *Cornus controversa* specific marker present
□ *Cornus alternifolia* specific marker present

**Fig 6. Presence and absence of *Cornus controversa* and *Cornus alternifolia* species-specific (diagnostic) RAPD and AFLP markers in putative hybrids and some individuals of C. *alternifolia* with undocumented origin.**

character (a distance from *C. alternifolia* of 0.49), and thus is interpreted as $F_1$-type. Two other plants (C1, C4) deviated from $F_1$ by only six and seven characters, respectively, (index = 0.43– 0.56) and were also classified as $F_1$-types. Manual calculations were confirmed using a computer program. Hybrid indices were calculated in the INTROGRESS package of the R environment only for specimens C1-C5, which were distinguished as the separate group in the aforementioned analyses, while the remaining specimens of *C. alternifolia* and *C. controversa* were used to train the program. All these samples turned out genotypically intermediate between the two groups of "pure" species, with hybrid indices ranging from 0.498 to 0.549.

## cpDNA

The amplification products obtained for *trn*F-*trn*L and for *psb*C-*trn*S regions of cpDNA had an expected length of about 950 bp and 1611 bp, respectively. Four of ten enzymes used in the analyses did not digest cpDNA regions. Most of the restriction products of amplified regions have shown no difference between dogwood accessions. The restriction patterns are identical to all restriction enzymes with the exception of *Dra*I, *Mse*I, and *Hinf*I for the *trn*F-*trn*L region and *Mbo*I for the *psb*C-*trn*S region. Following the digestion with these enzymes, differences in the length of restriction fragments were detected. These differences (from the four enzymes indicated above) are correlated and form distinct haplotypes, one characteristic for all samples of *C. alternifolia* studied and the second restricted to samples of *C. controversa* and hybrids (not shown). To sum up, the whole group of *C. controversa* specimens and hybrids had cpDNA haplotypes diagnostic for *C. controversa*. The complete exclusivity of the *C. controversa* chlorotypes in hybrids may indicate that maternally inherited cpDNAs were donated from this species, and therefore *C. alternifolia* is the potential paternal species and *C. controversa* is the maternal species.

## ITS and *PI*-like regions

Amplification with a new pair of species-specific ITS primers generally yielded fragments comparable to those described by White et al. [79]. However, a small difference in the length of PCR products between both species was observed. Compared to *C. alternifolia*, the PCR yielded fragments for *C. controversa* and putative hybrids that were smaller and non-differing in size. No differences were observed in the length of the products obtained in the case of amplifications of the *PI* loci. Electrophoresis of the denatured PCR products (single strand conformation polymorphism [SSCP]) resulted in a few major stable bands, different for each species. Their locations and distances between the upper and lower bands, especially for the ITS region, differed not only between the two *Cornus* species examined but also between particular individuals of the *C. alternifolia* group (not shown). These variations constituted simple and distinct SSCP patterns for individual species, allowing them to be distinguished. Furthermore, differences in size and conformation of DNA indicate the presence of a polymorphism, which may be further analyzed by sequencing.

Initially, sequence data for parental species available in GenBank: *C. controversa* ID: JF980315 (Oh et al. unpublished), ID: KP120055 [83], and C. *alternifolia* ID: DQ340526 [33] were compared. Despite the low level of ITS sequence variability and the sharing of the same sequence by both species as a result of their sequence alignment, single nucleotide polymorphism (SNP) markers were observed at 28 positions of the ITS 1 region. The PCR products

were cloned, and a number of individual recombinant clones were sequenced for each taxon. Following cloning, the sequences obtained for ITS regions were analyzed together with the sequences from GenBank. Finally nine unambiguous polymorphic SNP markers differentiating both species were selected. Sequences of hybrids (mostly identical to each other) indicated high similarity with the sequences of *C. controversa*. All five hybrids shared the majority (eight of the nine) of SNP markers differentiating both species and characteristic for *C. controversa*. Only one SNP marker characteristic for *C. alternifolia* was present in hybrid sequences. At the same time, the occurrence of four single nucleotides absent in the sequences of parental species and unique only for all hybrids was observed in their sequences.

The sequence alignment data for two specimens of the *C. alternifolia* group that were analyzed here have yielded surprising information, especially for A1 clones, which showed variation in 51 positions. Similarly, as in the case of hybrids, single nucleotides absent in the sequences of parental species were also observed in their sequences. However, in comparison to hybrids, their number was much higher. Furthermore, these clones had sequence data showing a *C. alternifolia* or *C. controversa* type, which is usually typical for hybrid origin. Some doubts were also raised by the results of the sequence analysis for plant A3 (individual shared markers unique for *C. controversa*), which showed variation in few positions. It seems to indicate the possibility that, apart from one individual (A2), from (A1-A3) *C. alternifolia* group, which has an almost identical sequence compared to the sequences obtained for our reference specimens as well published sequence, both other trees are not of pure origin.

Similarly, as described above (in the case of the ITS region), *Cornus PI*-like gene sequences of the parental species found in GenBank were aligned and compared with each other. Some species-specific markers for *C. controversa* (ID: EU447709) and *C. alternifolia* (ID: EU447710) were obtained as a result of sequence alignment between two parental species. Apart from one deletion in *C. alternifolia* (8 bp) and three in *C. controversa* (6, 2, and 1 bp), 19 SNP markers differentiating both species were detected in a fragment of about 550 bp. The PCR products obtained for parental species (studied here) and putative hybrids did not differentiate in size and seemed to be homogeneous. As done previously, they were cloned and a number of individual recombinant clones were sequenced for each taxon. Following cloning, sequences of *PI*-like genes were analyzed in different combinations of alignments to confirm hybrid origin/presence of introgression. As previously sequences of hybrids were almost identical to each other. Comparison of sequences for parental species with those obtained for the hybrids (clones) indicated higher similarity with the sequences of *C. alternifolia*. All five hybrids shared fourteen of the nineteen of SNP markers differentiating both species and characteristic for *C. alternifolia*. The characteristic for *C. controversa* type of the sequences or the occurrence of both types of sequences in hybrid clones also was observed. More importantly, in sequences of hybrids there were no deletions present in the *C. controversa* sequence type (they contain substitutions consistent with the sequences of *C. alternifolia*). At the same time they had substitution (8 bp) characteristic for the *C. controversa* type. Similarly as before single nucleotides absent in the sequences of parental species and unique only for hybrids also were observed in their sequences. Sequences obtained for *C. alternifolia* clones in general did not differ from the published sequences, although 2 SNPs specific for *C. controversa* were observed in the sequences of A1 clones.

## Discussion

Although hybridization is quite a common phenomenon in many organisms, particularly in plants, the occurrence of natural hybridization is not universal, according to Ellstrand et al. [84] is concentrated in a small fraction of plant families and genera. Molecular studies are

increasingly important in understanding the frequency and consequences of hybridization [85]. On the one hand, most putative hybrids that were identified via morphology were then subjected to molecular studies and were confirmed as hybrids [18]. On the other hand, molecular approaches have greatly increased the number of confirmed homoploid hybrids, which suggests that the frequency of this phenomenon was underestimated in the past, probably because it was much more difficult to detect [86].

Following some authors [84,85], "hybrid" refers to a hybrid type (or hybrid combination) derived from a unique combination of two parental species, and generally interspecific hybrids are commonly intermediate in their morphology between their parents. A number of examples, however, showed that distinguishing hybrids is not so simple, especially if there is no single morphological feature that can unambiguously distinguish the plants analyzed. Furthermore, defining the limits between the "typical" and "intermediate" individuals is often more or less arbitrary [87]. It should also be remembered that such morphologically intermediate forms, which are suspected to be hybrids, are regularly observed in natural mixed populations. In addition, a partial congruence between phenotypically and genetically intermediate individuals was found, suggesting that intermediate appearance does not necessarily mean hybridization [88].

In light of all this information, the first and main question that should be answered refers to the possibility of forming hybrids between the species studied here. Although they are described to be very closely related according to their phylogeny and morphology, the main reason that seems to suggest the low probability of their natural hybridization is the presence of geographic barriers between these two dogwoods. Such barriers do not exist in botanical collections where related species grow in proximity to each other. Since dogwoods are obligate outcrossers and considered self-sterile [89–91], the hybrids between closely planted individuals of different species might be expected. Among *Cornus* species with disjunctive geographical ranges, such examples occur naturally only in botanical gardens or are produced artificially (for details, see introduction).

The results of the preliminary morphological analysis of all *C. alternifolia* and *C. controversa* specimens examined proved that the average number of leaf lateral veins was in agreement with the available literature data concerning the particular species, while the shape of the lamina base was more variable in comparison with existing taxonomic descriptions [26,27,40]. Since the range of an average number of the leaf lateral veins overlaps and the shape of the lamina base is quite variable, these two morphological characters did not allow for the reliable taxonomic identification of five *Cornus* specimens of unknown origin (C1-C5) and may suggest their hybrid status.

SEM analyses of the adaxial and the abaxial leaf surfaces of both examined *Cornus* species confirmed some of the already known taxonomically important micromorphological characters; however, these analyses also yielded in finding few new not yet described features typical for the particular species. Some of these features are of taxonomic importance. The presence of pseudo-filiform trichomes on the adaxial leaf surface of both of the examined *Cornus* species was not reported before. According to the former results [41], *C. controversa* was described as one of several *Cornus* species with a glabrate vestiture on the adaxial leaf surface; however, the obtained results did not confirm such a description. The presence of the smooth wax layer on both sides of *C. controversa* leaves was recently reported [92]; however, the same results concerning the leaves of *C. alternifolia* and the presence of wax plates on the adaxial leaf surfaces of both examined *Cornus* species were not reported before. Besides these new findings, the sculpture of the adaxial leaf surface of both species under study was very similar and thus was not considered to be of taxonomic importance. In turn, the type of the sculpture of the abaxial lamina surface of two *Cornus* species under study appeared to be taxonomically important.

The different types of the trichomes observed on the abaxial lamina surface of *C. alternifolia* and *C. controversa* leaves confirmed the taxonomic importance of that character. On *C. alternifolia*, the single-armed pseudo-filiform trichomes were present on leaves, while two-armed flat-symmetrical appressed trichomes were observed on the *C. controversa* leaves. That is in agreement with most of the previously described results [26,27,41], with one exception. According to Rehder [40], the leaves of *C. alternifolia* are beneath appressed-pubescent, but that was not confirmed in the present study.

Besides the type of trichomes, the different form of the cuticular pattern present on the abaxial lamina surface of *C. alternifolia* and *C. controversa* leaves was species-specific. On *C. alternifolia* and *C. controversa* leaves, the papillose-striated or papillose-coronulate cuticular patterns were observed, respectively. These results are in agreement with Hardin and Murrell [41]. According to these authors, papillose- or coronulate-striated and filigree pattern are spectacular patterns of the abaxial leaf surface of many *Cornus* species. The filigree cuticle pattern on the abaxial side of *C. alba* and *C. sericea* leaves was recently described by Zieliński et al. [93]. Besides the main differences between the two cuticular patterns, which have been observed and described in the results of the present work, the following findings concerning the sculpture of the abaxial leaf surface were described here for the first time (1) the presence of a less expressive cuticular pattern on the abaxial lamina surface of *C. alternifolia* leaves compared to *C. controversa* leaves, (2) the occurrence of smaller, less numerous papillae that do not stand out against the surrounding sinuate cuticle striations, and (3) the absence or rare occurrence of the narrow cuticle interconnections joining the papillae on *C. alternifolia* leaves. We suggest to acknowledge these differences to be taxonomically important, however it is necessary to remember that the size of the papillae on the under surface of leaves can vary depending on the light [28].

Not all of the *C. controversa* specimens examined were characterized by the presence of the typical papillose-coronulate cuticular pattern on the abaxial leaf surface. Five of the examined *C. controversa* specimens, C1, C2, C3, C4, and C5 (the group of five specimens with the unknown origin), showed some clear distinctiveness compared to the papillose-coronulate cuticular pattern observed on the leaves of the *C. controversa* specimens with documented natural origin. The observed differences mainly include the shape of the papillae and their frequent appearance in each cuticle cell, the presence of thick and very intensive cuticle striations in places where there were no papillae, and the presence of both types of the trichomes on the abaxial lamina surface. The differences specified above indicate similarity with the cuticular pattern typical either for *C. alternifolia* or for *C. controversa* leaves, and thus they may suggest that the five *C. controversa* specimens of unknown origin (C1-C5) present characteristics typical for hybrids.

Based on morphological study, an analysis of naturally occurring putative homoploid hybrids between *C. controversa* and *C. alternifolia* using a molecular approach was performed. Five specimens of *Cornus* studied here were initially described as *C. controversa*, but since they showed traits intermediate between *C. alternifolia* and *C. controversa*, we proposed that they might be hybrids. Our preliminary studies using the two fingerprinting methods RAPD and AFLP seem to fully confirm this hypothesis. The RAPD markers had been used previously in a similar analysis of a *C. mas* × *C. officinalis* hybrid, and they successfully differentiated between both parent species and a putative hybrid [47]. AFLP analysis uses selective amplification of a subset of restriction enzyme-digested DNA fragments to generate a unique fingerprint of a particular genome [61]. Usually, in this method, the number of polymorphic fragments detected per reaction is much higher than that revealed by RAPD technology, but our experience indicates that the difference is not as big if RAPD products are also separated in acrylamide gels.

Because the initial analysis did not show significant differences in the tree topology constructed on the basis of the results of each method separately, all analyses were conducted on combined binary RAPD and AFLP data. Combining the results from both of the methods used here significantly increased the number of polymorphic fragments as well as unique bands in analysis. It is believed that the presence of unique markers from both parents identifies the presence of a hybrid; however, using few markers in the analysis causes the exact genetic classification of hybrid plants to be less certain [74]. A small number of markers also affects the identification of parental specimens. According to the authors, absence of markers from another species does not confidently establish that the plant being considered is "pure." They believe that "pureness" of parents is certain only within the limits of the number of markers that have been identified. More markers are being sought to increase confidence in the status of the parents. Soltis et al. [94], in turn, noted that the number of polymorphic markers used in analysis influenced the position of hybrids in the tree. According to these authors, a small number might predict that a diploid hybrid may group with either parent, and as the number of polymorphic sites increases, the hybrid could appear in a completely novel position, well removed from both parents.

Both RAPD and AFLP methods detected a high and comparable level of polymorphism among the 22 genotypes representing two species of *Cornus* and putative hybrids between them. Polymorphism in the *C. controversa* group is slightly higher than that observed in the *C. alternifolia* group; however, the difference is small (85% versus 83.5%). Among all polymorphic products scored for *C. controversa* and *C. alternifolia* in both methods, 388 DNA bands differentiated parental species Apart from these, some bands unique for putative hybrids were also present in their profiles. According to some authors, these non-parental bands may be generated from the recombination and mutation in meiosis processing during hybridization [95,96] and may also be created by heteroduplex formation [97]. Such hybrid-specific bands (not seen in parental genomes) are useful for identification of specific hybrids. More recent results of the study by Zhao et al. [98] suggest that more genetic diversity and new variation could be captured by crossing breeding. It is not clear whether these novel bands may be responsible to new genes associated to important traits because it is still difficult to address particular mechanisms that help us understand the chromosomal or genomic rearrangements in response to novel bands [98].

All analyses based on combined RAPD/AFLP data confirmed separate and intermediate status of the specimens classified preliminarily as the putative hybrids: they were repeatedly located between parental species groups. Only in UPGMA did the putative hybrids create a distinct subgroup in the clade of *C. controversa*. The same pattern also appeared in the model-based method implemented in STRUCTURE, where the combination of two clusters typical to both parental species was present in all putative hybrid specimens. For the higher number of clusters (K = 3), accessions C1-C5 were nearly completely attributed to a third "mixed ancestry" cluster. Moreover, this cluster also had quite a distinct share in individuals A1 and A3, which might indicate the repeated backcrossing of an interspecific hybrid with *C. alternifolia*. In both these specimens, the information about their provenance was lacking; therefore, it cannot be excluded that these specimens could originate as a result of backcrossing in a garden collection. Although NJ, PCoA, and STRUCTURE analysis clustered the accessions similarly to a great extent, the advantage of the latter was that it enabled us to recognize the admixture and pure ancestry of accessions without prior reference samples of "pure" genotypes.

Hybridization can also be examined based on the distribution of marker bands under various established criteria that determine private-marker or shared-band status. Finally, the distinctiveness of the parental species was identified by species specific (species-diagnostic) bands that are always present in the profiles of each species. They were identified to determine the

level of pairwise genetic relationships and to reveal potential hybridization and introgression direction.

Based on 95 unambiguous marker loci for putative hybrids C1-C5 and the three *C. alternifolia* specimens A1-A3 (eight plants in the majority, of unknown origin), the hybrid index was calculated. Hybrid index estimates indicated that all hybrids were the first filial generation of offspring ($F_1$) of both parental species. Although all hybrids seemed to be approximately equally related to each parent, they were found to be heterogeneous in their collection of parental markers. The heterogeneity in these plants may result from segregation of parental markers that are possibly heterozygous. At the same time, our data also revealed evidence of hybridization not suspected from morphology. Three plants (C1, C2, and C5) initially considered to be "pure" *C. controversa* based on their morphological traits were found to be $F_1$ or nearly so in the genetic analysis. This indicates that morphology may not necessarily reflect the genetic contribution of the two parental species. Noteworthy is the fact that one of the specimens within the group of *C. alternifolia* (A3) shared the subset of bands (four) that was diagnostic to *C. controversa*. The presence of such markers can suggest introgressive hybridization between *C. controversa* and *C. alternifolia* in the past. According to Kirk et al. [99], putative hybrids were considered to be confirmed hybrids if they possessed at least one diagnostic marker from each parental species or if they possessed at least one diagnostic marker from one parental species and were missing at least one uniform diagnostic marker from the same parental species. Following this definition, our results may raise doubts about the *C. alternifolia* individual described above. Based only on the value of hybrid index (0.15), this plant (A3), similar to the A1 individual (0.16), is interpreted here as almost "pure".

Some authors, such as Hegarty and Hiscock [23] or Schwarzbach and Rieseberg [100], point to the importance of not relying on a single-marker system when studying complex genetic events like speciation/hybridization. According to these authors, such studies require the accumulation of evidence from multiple sources before definitive answers can be given. There are at least several reasons why extending the study will give greater certainty in the assessment of hybridization or introgression. Many hybrids will not be encountered, especially when species can be thoroughly examined throughout their ranges. Furthermore, ongoing cases of hybridization will be detected much easier than cases that occurred in the recent or distant past, which is probably the case with the hybrids analyzed here. According to Rieseberg and Wendel [101], the signal of hybridization may be diluted over time by different factors such as mutations in the introgressed alleles, genetic drift, and natural selection against introgressed alleles, all of which make hybrids more difficult to identify. There are also other more complicated cases of hybridization that are unlikely to be suspected or described if the parents are not immediately obvious. Finally, in some groups with substantial barriers to interspecific contact, recent molecular work has identified "cryptic" hybrids, suggesting that overall levels of hybridization may be higher than currently thought [18,102].

With these comments in mind, we have to refer to some limitations in our research such as a limited number of tested individuals (both species are not native to our country) and lack of data for some of them. Unfortunately, the studied species are perennial plants and their history begins before the World War II; therefore, the archives do not have full information from that period. We do not know if the hybrids are progenies of some plants under study, but such a possibility cannot be excluded. Some of the mentioned specimens are connected by the date of planting, which may indicate a common place of origin, but there is no evidence for it. Some doubts are also raised by the results obtained for the group A1-A3 *C. alternifolia*, although they are grouped within "pure" specimens, a certain distinctiveness of individuals A1 and A3 is clearly visible. To be certain, we decided to add additional analyses using other markers to confirm the hybrid origin/presence of introgression in the specimens studied here.

Uniparental inheritance and haploid character, cpDNA reveals half of the parentage in plants of hybrid origin (generally, the seed parent in angiosperms). Therefore, the possibility should be taken into account that cpDNA analysis of hybrid plants may incorrectly identify them as belonging to a clade of one of the two parents [52]. In our study, differences in the length of restriction fragments formed two distinct haplotypes, one characteristic for *C. alternifolia* and the second restricted to *C. controversa* and hybrids. Such exclusivity of the chlorotypes in hybrids may indicate that maternal inherited cpDNAs were donated from *C. controversa*.

Two nuclear genes (rDNA and *PI*-like gene) were also adopted as tools in this study. ITS sequence data have been extensively documented to provide insights into phylogenetic history and historical introgression in different plants, including Cornaceae [33,103,104]. They have also been successfully used to document the hybrid origin of several species [105–107].

The differences in size and pattern of SSCP banding between PCR products of parental species and hybrids were observed. Differences were also noticed between two specimens (A1, A3) and the rest of the *C. alternifolia* group indicating the presence of polymorphism. Amplification products were sequenced, and sequence data were compared to each other. The results of the comparative analysis of the ITS sequences confirm the fact that the studied species are closely related, which is in line with the analyses resulting from morphology and cytology. Apart from the observed polymorphism (SNP), the diversity of ITS sequences within the studied group is small. The morphological diversity seems to contrast with the low levels of ITS sequence variability and the sharing of the same sequence by different species. However, such examples where the groups of morphologically differentiated species were found to have identical ITS sequences have already been described. According to Emshwiller and Doyle [105], the reasons for such sequence similarity may be very different. They suggested low levels of divergence, the wide spread of the interspecific gene flow, or the possibility of the existence of some specimens with morphologies intermediate between species, which could be hybrids. On the other hand, we observed differences in the sequences of accessions that were identified as being from the same species (different sequence types usually differing at a single nucleotide position). Emshwiller and Doyle [105] suggested that this could be due either to intraspecific polymorphism or, in some cases, to the taxonomic uncertainties. According to some authors, such examples, polymorphic positions, are found frequently in published rDNA data sets. What is more, Álvarez and Wendel [103] and Small et al. [52] think that such variation is the norm for rDNA rather than the exception.

Sequences obtained for hybrids, with one exception, indicate a closer relationship with one of the parents. They mostly represent sequence type *C. controversa*. Such results where sequencing revealed even only a single rDNA sequence type characteristic of one genome but not both have already been reported [103]. For example, Yang et al. [108] reported that there was evidence to suggest that concerted evolution within different repeats of ITS regions can quickly eliminate one parental and completely homogenize to the other parental lineage (to the maternal lineage). It could also be a probable explanation in our case.

In the case of *C. alternifolia* accessions, detailed analysis of sequence data for A1 clones revealed the typical for hybrid origin presence of two variants of sequence of the *C. alternifolia* or *C. controversa* type. Our data also indicated that individual of *C. alternifolia* described here as A3, also is not of pure origin. The taxonomic position of A1 and A3 is still not clear. From the sequencing and fingerprinting results, we can only conclude that these specimens could originate as a result of backcrossing.

Noteworthy is the presence of nucleotides absent in any of the parents, both in the case of hybrids and especially *C. alternifolia* clones (A1 and A3), which showed such variation in over fifty positions. It is not easy to explain its occurrence, but it should be noted that the strength

of concerted evolution is very different across units, arrays, and taxa yielding different rDNA types within specimens. Generally, analysis of rDNA sequences, especially in the case of unrecognized hybrids, is often very difficult. Apart from the simplest explanation connected with the presence of hybridization/introgression, there are other different possible reasons for the presence of sequence types in particular species. It is possible that during PCR the repeat type that is already present in greater numbers may be favored, or differences result from the sequencing method (cloned versus directly sequenced PCR products) [109]. On the other hand, sequence polymorphism may go undetected in sequencing, or the strongest peaks (or bands) may be scored as the correct bases [94]. Such problems become exacerbated in hybrid taxa in which rDNA loci are expected to exist on different chromosomes donated by different parents.

Phylogenies based on ITS data may be incongruent with those based on other markers due to the various mechanisms that influence ITS. Feliner and Rosello [50] also point out the fact that analyses can be misleading either due to complete homogenization of ITS in unsuspected hybrids, partial homogenization, or segregational loss in later-generation hybrids when examining only one locus (ITS). That's why we decided to add to the analysis an alternative region of a *Cornus PI*-like gene containing a useful phylogenetic signal at the species level. Theoretically, such low-copy nuclear genes offer advantageous properties because they are expected to contain more variable sites, the presence of multiple independent loci and biparental inheritance. Alignment of *PI*-like sequences of parental species with that obtained for the cloned region of hybrids fully confirmed the hybrid origin of all five specimens C1-C5. Sequence data provided some evidence for the contribution of both parents to the genomes of putative hybrids. The data obtained also confirmed that at least two *C. alternifolia* accessions are not free of introgression. Moreover, these results concluded that the use of the SNPs of ITS or other low-copy nuclear genes may be considered an accurate tool to characterize hybrid origin. In summary, we found that molecular markers (RAPD/AFLP), sequences, and morphological leaf traits are highly coincident and support the phenomenon of hybridization between *C. controversa* and *C. alternifolia*. We confirmed the hybrid origin of all five putative hybrids. Moreover, all molecular analyses indicate that they are $F_1$ or $F_1$-type hybrids. Our results also showed that two individuals in the *C. alternifolia* group, phenotypically non-divergent A1 and A3, are not free of introgression. They are probably backcrosses toward *C. alternifolia*. Although based on the RAPD/AFLP they were included in the group of almost "pure" species, the analysis of the sequence of A1 clones showed a typical for the hybrid presence of the sequence of both parent species. In the case of the A3 individual, repeated backcrossing into the *C. alternifolia* lineage might restore its nearly "pure" genome (this process generates a recombinant ribosomal DNA), which will explain its ITS sequence identity. Aguilar et al. [110] also showed that backcrosses showed almost complete homogenization in the direction of the recurrent parental species. In turn, Wendel et al. [111] described a similar example as cryptic intergenomic introgression between species, which, like *Cornus* species in this study, now occupy different hemispheres. Furthermore, molecular data seem to point not only to unidirectional introgression toward *C. controversa*, which is evidenced by the presence of the hybrids C1-C5. Bidirectional introgression also cannot be excluded since the presence of markers specific for *C. controversa* in the profiles of *C. alternifolia* (A3) was observed.

## Supporting information

**S1 Fig. Geographical distribution of *Cornus alternifolia* and *C. controversa*.** The compilation based on Thompson et al. (1999) and POWO (2019).
(PDF)

**S2 Fig. SEM micrographs of the adaxial leaf surface.** *C. alternifolia* (a–specimen A9; b–A4; c, d–A2; e–A5; f–A1) and *C. controversa* (g, h–specimen C21; i–C20; j–C6; k, p–C3; l–C4; m–C5; n–C1, o–C2). Hybrid specimens C1-C5 (k-p), show the reticulate microoornamentation pattern with parallel (f, k, m) or wavy (a-e, g, i, j, l, p) cuticle striations, raised (a, c, e, k, l, o, p) or flat (b, d, f, h, j, m, n) anticlinal cell walls and straight pseudo-filiform trichomes (e, f, n, o, p). Magnitude 915–1200; specimen symbols as in Table 1.
(PDF)

**S3 Fig. UPGMA dendrogram of *Cornus alternifolia* and *C. controversa* accessions, and *C. macrophylla* as an outgroup, based on Dice similarities calculated from RAPD and AFLP combined binary matrices.** Accessions codes as in Table 1; numbers at the nodes indicate bootstrap support.
(PDF)

**S4 Fig. An unrooted neighbour-joining (NJ) tree of studied *Cornus alternifolia* and *C. controversa* accessions based on Dice distances calculated from RAPD and AFLP combined binary matrices.** Accessions codes as in Table 1; numbers at the nodes indicate bootstrap support.
(PDF)

**S1 Table. Comparison of RAPD and AFLP profiles for *Cornus controversa (C.c.)*, *Cornus alternifolia (C.a.)*, putative hybrids (H) and *Cornus macrophylla (C.m.)*.** The number of DNA bands unique to each parental species and *C. macrophylla* are compared with the number of species-specific (diagnostic) bands.
(PDF)

## Acknowledgments

We would like to express our gratitude to all the colleagues and friends who helped us to prepare this paper, namely to Professor Jerzy Zieliński (Institute of Dendrology in Kórnik), for bringing our attention to the problem, to Piotr Banaszczak (Director of the Rogów Arboretum) and Kinga Nowak (Director of the Kórnik Arboretum), for their help in material collection, and to Wojciech Klimko (Department of Entomology and Environmental Protection, Poznań University of Life Sciences), for his technical assistance in preparation of SEM figures.

## Author Contributions

**Conceptualization:** Barbara Gawrońska, Maria Morozowska, Piotr Kosiński.

**Data curation:** Barbara Gawrońska, Maria Morozowska, Piotr Kosiński.

**Formal analysis:** Barbara Gawrońska, Maria Morozowska, Piotr Kosiński.

**Funding acquisition:** Barbara Gawrońska, Maria Morozowska, Piotr Kosiński.

**Investigation:** Barbara Gawrońska, Maria Morozowska, Katarzyna Nuc, Piotr Kosiński.

**Methodology:** Barbara Gawrońska, Maria Morozowska, Piotr Kosiński, Ryszard Słomski.

**Project administration:** Barbara Gawrońska, Maria Morozowska, Piotr Kosiński, Ryszard Słomski.

**Resources:** Barbara Gawrońska, Maria Morozowska, Piotr Kosiński.

**Software:** Barbara Gawrońska, Maria Morozowska, Piotr Kosiński.

**Supervision:** Barbara Gawrońska, Maria Morozowska, Piotr Kosiński.

**Validation:** Barbara Gawrońska, Maria Morozowska, Piotr Kosiński.

**Visualization:** Barbara Gawrońska, Maria Morozowska, Piotr Kosiński.

**Writing – original draft:** Barbara Gawrońska, Maria Morozowska, Piotr Kosiński.

**Writing – review & editing:** Barbara Gawrońska, Maria Morozowska, Piotr Kosiński, Ryszard Słomski.

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
