## [Decision Letter · Decision Letter 0]

25 Nov 2019

PONE-D-19-30939

What nature had separated, and human has joined together: about a spontaneous hybridization between two allopatric dogwood species (Cornus controversa and C. alternifolia)

PLOS ONE

Dear Mrs. Gawrońska,

Thank you for submitting your manuscript to PLOS ONE. After careful consideration, we invite you to submit a revised version of the manuscript that addresses the points raised by our two reviewers -- see below.

We would appreciate receiving your revised manuscript by Jan 09 2020 11:59PM. To enhance the reproducibility of your results, we recommend that if applicable you deposit your laboratory protocols in protocols.io, where a protocol can be assigned its own identifier (DOI) such that it can be cited independently in the future. For instructions see: http://journals.plos.org/plosone/s/submission-guidelines#loc-laboratory-protocols

We look forward to receiving your revised manuscript.

Kind regards,

Pawel Michalak

Academic Editor

PLOS ONE

Journal Requirements:

Reviewers' comments:

Reviewer's Responses to Questions

**Comments to the Author**

1. Is the manuscript technically sound, and do the data support the conclusions?

Reviewer #1: Yes

Reviewer #2: Yes

2. Has the statistical analysis been performed appropriately and rigorously? 

Reviewer #1: N/A

Reviewer #2: Yes

3. Have the authors made all data underlying the findings in their manuscript fully available?

Reviewer #1: Yes

Reviewer #2: Yes

4. Is the manuscript presented in an intelligible fashion and written in standard English?

Reviewer #1: Yes

Reviewer #2: Yes

5. Review Comments to the Author

Reviewer #1: In 'What nature had separated, and human has joined together: about a spontaneous hybridization between two allopatric dogwood species (Cornus controversa and C. alternifolia)', the authors explored a possible hybridization between two allopatric species, Cornus controversa and Cornus alternifolia via various approaches, including scanning electron microscope, combination of RAPD and AFLP, and ITS sequencing. Based on the data from those methods, the authors confirmed the hybridization, and inferred the hybridization direction from intergenic spacer sequences from chloroplast. The authors also found introgression from hybrid to C. controversa. This is an interesting study. The manuscript is generally well written and structured, the conclusion is supported by the data. Even though the number of samples/specimens included in this study is limited, the discussion is thorough and comprehensive.

Few comments:

1) In the abstract, avoid using acronyms.

2) Page 7, section "Plant material", it is not clear to me that how many tree samples are those 30 specimens collected from. If the number is less than 30, then why collecting multiple specimens from same sample?

3) Table 1, what does accession number mean? Does same accession number indicate the same tree/sample?

4) It would be interesting if the authors can include samples of the two speices from their original places and compare them with the current samples.

Reviewer #2: This study presents results of primary scientific research and to the reviewer’s knowledge are not reported and published elsewhere. The experiments, statistics and other analyses are performed to a high technical standard and the justification and rationale are outlined clearly throughout the paper. The conclusions are presented in a clear and sequential format supported by experimental data. The written English of this paper is not in question. Regarding ethics of experimentation and research, plant material here was collected and/or received in an ethical manner, and relevant acknowledgements were made. All necessary data has been made available either through supplemental material or through publicly accessible platforms.

Regarding minor revision:

Material

Table 1

The alignment of location and individual is not clear due to some table formatting issues.

A9 location information should be corrected to Fargo, North Dakota

Methods

For micromorphological observations, how many samples were used? This is listed for macromorphological analysis. Was it the same for both? If so, then please clarify.

6. PLOS authors have the option to publish the peer review history of their article (what does this mean?). If published, this will include your full peer review and any attached files.

Reviewer #1: No

Reviewer #2: No

---

## [Author Response · Author response to Decision Letter 0]

5 Dec 2019

Responses to the Editor’s and Reviewers' Comments

We put forward a corrected version of our manuscript entitled “What nature separated, and human joined together: about a spontaneous hybridization between two allopatric dogwood species (Cornus controversa and C. alternifolia)”. We would like to thank the Reviewers and the Editor for the effort and time put into the review of the manuscript and for the interest in our work and constructive comments that will greatly improve the manuscript. As indicated below, we have checked all the general and specific comments provided by the Editor and Reviewers. Necessary corrections and supplementations have been introduced to the revised version of our manuscript.

Reviewer #1 comments:

1) In the abstract, avoid using acronyms.

The full names of the used acronyms were introduced into the abstract. 

2) Page 7, section "Plant material", it is not clear to me that how many tree samples are those 30 specimens collected from. If the number is less than 30, then why collecting multiple specimens from same sample?

Thank you very much for the careful reading of our manuscript. Each sample was taken from a different individual/specimen. As result, we collected nine samples of C. alternifolia, 21 samples of C. controversa (including putative hybrids), and additionally two samples of C. macrophylla (used as an out-group in some analyses). This last piece of information (concerning C. macrophylla) was missing in the description of the study material. We added it to the text.

3) Table 1, what does accession number mean? Does same accession number indicate the same tree/sample?

As we have mentioned previously, each sample represents different individual. Accession numbers are inventory codes of specimens in given collections. We change caption of the column from ‘Accession number’ to ‘Inventory code’. 

4) It would be interesting if the authors can include samples of the two species from their original places and compare them with the current samples.

Since both species under study are not native to Europe we tried to do our best in gathering fresh plant material for molecular and micromorphological analyses from specimens either growing in their original places or reproduced from such individuals. The examined by us plant material includes several samples representing specimens of documented natural origin. These are individuals which were reproduced from seeds collected from C. alternifolia and C. controversa specimens of natural origin (e.g. specimens A2, A5, C9, C16, C17). Additionally, the herbarium material examined under SEM was also obtained from the specimens which collected in natural sites (e.g. specimens A9, C9, C18). 

Reviewer #2 comments: 

Material

Table 1

The alignment of location and individual is not clear due to some table formatting issues.

The table layout has been improved. 

A9 location information should be corrected to Fargo, North Dakota.

Thank you very much for the careful reading of our manuscript. We corrected this flaw.

Methods

For micromorphological observations, how many samples were used? This is listed for macromorphological analysis. Was it the same for both? If so, then please clarify.

The micromorphological observations were done on a sample of 5-10 leaves per individual, depending on the material availability. This correction was introduced into the text.

---

## [Editor Report · Decision Letter 1]

11 Dec 2019

What nature separated, and human joined together: about a spontaneous hybridization between two allopatric dogwood species (Cornus controversa and C. alternifolia)

PONE-D-19-30939R1

Dear Dr. Gawrońska,

We are pleased to inform you that your manuscript has been judged scientifically suitable for publication and will be formally accepted for publication once it complies with all outstanding technical requirements.

With kind regards,

Pawel Michalak

Academic Editor

PLOS ONE
---

## [Editor Report · Acceptance letter]

13 Dec 2019

PONE-D-19-30939R1 

What nature separated, and human joined together: about a spontaneous hybridization between two allopatric dogwood species (*Cornus controversa* and C*. alternifolia*) 

Dear Dr. Gawrońska:

I am pleased to inform you that your manuscript has been deemed suitable for publication in PLOS ONE. Congratulations! Your manuscript is now with our production department. 

With kind regards,

on behalf of

Dr. Pawel Michalak 

Academic Editor

PLOS ONE